# Inferring Bacterial Community Interactions and Functionalities Associated with Osteopenia and Osteoporosis in Taiwanese Postmenopausal Women

**DOI:** 10.3390/microorganisms11020234

**Published:** 2023-01-17

**Authors:** Yi-Jie Kuo, Chia-Jung Chen, Bashir Hussain, Hsin-Chi Tsai, Gwo-Jong Hsu, Jung-Sheng Chen, Aslia Asif, Cheng-Wei Fan, Bing-Mu Hsu

**Affiliations:** 1Department of Orthopedic Surgery, Wan Fang Hospital, Taipei Medical University, Taipei 116, Taiwan; 2Department of Chinese Medicine, Dalin Tzu Chi Hospital, The Buddhist Tzu Chi Medical Foundation, Chiayi 622, Taiwan; 3Department of Earth and Environmental Sciences, National Chung Cheng University, Chiayi 621, Taiwan; 4Department of Biomedical Sciences, National Chung Cheng University, Chiayi 621, Taiwan; 5Department of Psychiatry, School of Medicine, Tzu Chi University, Hualien 970, Taiwan; 6Department of Psychiatry, Tzu-Chi General Hospital, Hualien 970, Taiwan; 7Division of Infectious Disease, Department of Internal Medicine, Chia-Yi Christian Hospital, Chiayi 621, Taiwan; 8Department of Medical Research, E-Da Hospital, I-Shou University, Kaohsiung 824, Taiwan; 9Doctoral Program in Science, Technology, Environment and Mathematics, National Chung Cheng University, Chiayi 621, Taiwan

**Keywords:** osteopenia, osteoporosis, gut microbiota, 16S rRNA sequencing, functional prediction, driver bacteria

## Abstract

Growing evidence suggests that the gut microbiota and their metabolites are associated with bone homeostasis and fragility. However, this association is limited to microbial taxonomic differences. This study aimed to explore whether gut bacterial community associations, composition, and functions are associated with osteopenia and osteoporosis. We compared the gut bacterial community composition and interactions of healthy postmenopausal women with normal bone density (*n* = 8) with those of postmenopausal women with osteopenia (*n* = 18) and osteoporosis (*n* = 21) through 16S rRNA sequencing coupled with network biology and statistical analyses. The results of this study showed reduced alpha diversity in patients with osteoporosis, followed by that in patients with osteopenia, then in healthy controls. Taxonomic analysis revealed that significantly enriched bacterial genera with higher abundance was observed in patients with osteoporosis and osteopenia than in healthy subjects. Additionally, a co-occurrence network revealed that, compared to healthy controls, bacterial interactions were higher in patients with osteoporosis, followed by those with osteopenia. Further, NetShift analysis showed that a higher number of bacteria drove changes in the microbial community structure of patients with osteoporosis than osteopenia. Correlation analysis revealed that most of these driver bacteria had a significant positive relationship with several significant metabolic pathways. Further, ordination analysis revealed that height and T-score were the primary variables influencing the gut microbial community structure. Taken together, this study evaluated that microbial community interaction is more important than the taxonomic differences in knowing the critical role of gut microbiota in postmenopausal women associated with osteopenia and osteoporosis. Additionally, the significantly enriched bacteria and functional pathways might be potential biomarkers for the prognosis and treatment of postmenopausal women with osteopenia and osteoporosis.

## 1. Introduction

Osteopenia is a metabolic bone disorder characterized by reduced bone mineral density (BMD), a common precursor of osteoporosis, which is considered a significant global health issue among elderly individuals, leading to an increased risk of bone fragility [1,2,3,4]. Osteoporosis in postmenopausal women is typically associated with estrogen deficiency, and it is estimated to affect more than 200 million women globally, leading to significant medical and economic burdens worldwide [5,6]. Bone resorption and formation are tightly linked to the balance between osteoclasts and osteoblasts [7,8]. Several factors, including estrogen deficiency, aging, smoking, and continuous calcium loss, have been reported to be responsible for disturbing this tightly coupled process, eventually leading to osteoporosis in postmenopausal women [1,9,10]. Growing evidence suggests that gut microbes play a vital role in bone health by transporting and absorbing the necessary nutrients [2]. Previous studies have reported that gut microbiota can regulate bone health [11] through the intervention of the immune system [12], intestinal calcium absorption [13], and neurotransmitter release [14]. However, the mechanistic understanding of the effects of these microorganisms on particular physiological processes is still in its infancy.

The human gut microbiota consists of vast and complex microbial communities comprising 10^13^–10^14^ microorganisms and approximately 100–500 different species per individual [15,16]. These microbial cells remain in complex metabolic and ecological associations that ultimately influence the functional and taxonomic profiles of the microbial community and host health [17]. To infer the microbial community, it is important not only to focus on microbial abundance but also to enumerate the changes observed among the microbial interactions and associations [18]. However, the majority of earlier studies focused only on taxonomic differences between the gut microbial communities of patients with osteopenia and osteoporosis in comparison with healthy controls [1,11,19,20]. Although such comparisons are valuable, profiling of the microbial community is not sufficient to elucidate the underlying microbial interactions and their effects on both the microbial ecosystem and host health [17,18]. As a result, our understanding of how the gut microbial community is associated with osteopenia and osteoporosis remains unclear.

Recently, growing evidence has suggested that gut microbial metabolites could also induce bone remodeling in both animal and human models [21,22]. According to Ling et al. (2020) and Yan et al. (2016), low BMD in osteopenia might be associated with several microbial functional pathways, including membrane transport and lipopolysaccharide biosynthesis. However, few studies have evaluated the association or relationship between the predicted microbial functions and bone health associated with humans [20,23]. Owing to the lack of sufficient knowledge about the associations of gut microbial communities and their functions pertaining to bone health, it is not yet clear whether the onset of osteoporosis occurs due to disruption of the gut microbiota community. Therefore, inferring the gut microbial taxonomic compositional changes and microbial interactions associated with bone health is essential to categorize the underpinning microbial taxa and investigate their functions, which will help researchers understand the association between the gut microbial community and bone mass disorders. Hence, this study aimed to explore the gut bacterial community associations, composition, and functional changes associated with osteopenia and osteoporosis in postmenopausal Taiwanese women using 16S rRNA sequencing coupled with network biology and statistical analysis.

## 2. Materials and Methods

### 2.1. Characterization of Healthy Participants and Patients

A total of 47 postmenopausal women were enrolled in the analysis of fecal samples. Qualifying patients were postmenopausal women, older than 50 years, who have completed BMD assessment using dual-energy X-ray absorptiometry in the last 3 months. Participants were excluded if they had lower BMD resulting from secondary factors, including underlying endocrine, autoimmune, gastrointestinal, hepatic, and nutritional disorders, or drug-induced bone loss (e.g., steroids and proton pump inhibitors). Participants who are alcoholics or drug addicts, who have been on antibiotics or probiotics for the past 3 months, have symptoms of constipation for more than one week or diarrhea for more than 2 weeks, or have a surgical history of the gastrointestinal tract were also excluded.

All participants underwent DXA assessment (Prodigy, GEHC Lunar, Madison, WI, USA) to analyze the BMD of the lumbar spine and bilateral proximal femurs to determine the T-score of all parts of interest. The T-score was recorded as the lowest T-score of the bilateral hip and spine regions, reflecting a comparison of BMD with that of healthy young adults (aged 20 years) [24]. A T-score higher than −1, between −1 and −2.5, and lower than −2.5 indicates normal bone density, osteopenia, and osteoporosis, respectively. Overall, this study enrolled 8 healthy subjects with normal BMD, 18 patients with osteopenia, and 21 patients with osteoporosis. The characteristics of the enrolled participants are shown in Appendix A.

### 2.2. Sample Collection and Genomic DNA Extraction

The fecal specimens of healthy subjects and postmenopausal women with osteopenia and osteoporosis were collected in a sterile stool box and cryopreserved at the Taipei Municipal Wanfang Hospital, Taiwan. Samples were then transported according to biosafety procedures and under controlled temperature conditions to the laboratory at the National Chung Cheng University, Taiwan. Fecal gDNA was extracted from a 200 mg stool sample using a QIAamp DNA Stool Mini Kit (QIAGEN, Hilden, Germany) according to the manufacturer’s instructions. Additionally, a bead-beating step was performed using a previously designed protocol [25]. In brief, 250 µL of the stool sample was put in a 2 mL sterilized tube; subsequently, 1.2 mL ASL lysis buffer and 0.3 g sterile 0.1 mm zirconia beads (BioSpec, Bartlesville, OK, USA) were added and vortex-mixed for 2 min. The samples were heated for 15 min at 95 °C and then homogenized using a Qiagen TissueLyser II. After treatment with an InhibitEX Tablet, 350 µL of the supernatant was transferred to another tube to perform the subsequent purification steps using a QIAcube system.

The purity and concentration of the extracted gDNA were determined using a Nanodrop 2000 spectrophotometer (Thermo Fisher Scientific Inc., Wilmington, DE, USA) at 230–280 nm. The quality of the gDNA was examined using gel electrophoresis (1.5% gel in Tris-acetate ethylenediaminetetraacetic acid buffer) at 110 V for 30 min. The DNA bands were visualized under ultraviolet light. The purified gDNA was stored at −20 °C until further analysis.

### 2.3. Sequencing, Library Construction, and Microbial Community Analysis

The gDNA purified from healthy subjects and patients was used to amplify the V3–V4 hypervariable regions of 16S rRNA using the paired-end Illumina MiSeq platform (Illumina Inc., San Diego, CA, USA). The targeted sequence was amplified using PCR with forward and reverse primers, as previously described by Fang et al. [16]. The primer sequences and PCR amplification conditions were set as described by Hussain et al. (2021). The quality and quantity of the amplified DNA were examined using standard quality checks, as mentioned in the gDNA extraction section. Next, sequencing was performed using 20 µL amplicons from each sample following the pair-end method with the MiSeq Illumina platform (Illumina Inc., San Diego, CA, USA) at the National Yang-Ming University Genome Research Center, Taiwan. The Illumina Nextera XT kit was used to ligate the sequence adapters and index, according to the manufacturer’s instructions. Additionally, sequence data ligation of forward and reverse reads was performed using the CLC bio plate form (Genomic Workbench v.8.5). FASTA files were generated as described by Huang et al. [26]. We used QIIME2 as a sequence analysis tool to further analyze the FASTA files. After merging and removing duplicate sequences using DADA2, high-quality sequences were clustered into amplicon sequence variants (ASVs) at a 97% similarity index. To analyze bacterial diversity, the first rarefaction was performed at the lowest sequence depth, and alpha diversity was measured based on experimental groups. Additionally, beta diversity was measured based on the Bray–Curtis index, followed by the permutational multivariate analysis of variance (PERMANOVA) method. The relative abundance of microbes at the phylum and genus levels in each sample was determined using the QIIME2 view. Furthermore, the significant difference in the relative abundance at the genus level among the experimental groups was analyzed using statistical analysis of taxonomic and functional profiles based on a two-tailed Welch t-test (*p* < 0.05) followed by the Benjamini–Hochberg method to control the false discovery rate. Pearson analysis at the genus level was performed to analyze co-occurrence correlation based on two-sided pseudo *p* < 0.05, using the MetagenoNet tool [27]. Furthermore, NetShift analysis [18] was conducted to detect the driver bacteria at the genus level among the experimental groups based on the networks generated by Pearson analysis.

### 2.4. Functional Profiling Analysis among the Experimental Groups Based on 16S rRNA Gene Signatures

Bacterial functional prediction of experimental groups was performed using the phylogenetic investigation of community reconstruction of the Unobserved States (PICRUSt2) pipeline (https://github.com/picrust/picrust2; accessed on 1 June 2022) based on the Metacyc database (https://metacyc.org/; accessed on 1 June 2022). The representative sequence and denoised ASV abundance table were used as inputs for PICRUSt2. All ASVs with the nearest-sequenced taxon index (cutoff value of >2) were removed by default to reliably annotate metabolic functions using the KEGG reference database, as previously described by Douglas et al. (2020) [28]. Finally, pathways that were statistically significant based on a two-tailed Welch t-test (*p* < 0.05) with Benjamini–Hochberg FDR were selected to evaluate the differences between experimental groups. Pearson correlation analysis was performed to evaluate the significant correlations between bacterial taxa and potential functional prediction considering *p* ranging from 0.01–0.05 using IBM SPSS Statistics 24 (IBM, Armonk, North Castle, NY, USA) software to evaluate the significant correlations between bacterial taxa and potential functional prediction.

## 3. Results

### 3.1. Sequencing and ASV Analysis among the Experimental Groups (Normal Control, Osteopenia, and Osteoporosis) Based on 16S rRNA Gene Signatures

The 16S rRNA gene sequence based on the V3 and V4 regions was analyzed to compare the differences in bacterial community diversity and abundance in the normal control, osteopenia, and osteoporosis groups. A total of 615,572 sequences were obtained from 47 samples belonging to three groups: normal control, osteopenia, and osteoporosis. Among them, there were 322,786 good quality sequence reads for downstream analysis after chimeric sequence removal and quality filtering by considering the 97% cut-off range. Rarefaction of these obtained sequences was performed at the lowest sequence depth to compare bacterial diversity among the normal control, osteopenia, and osteoporosis groups, as shown in Appendix A. The rarefaction analysis showed a high diversity associated with the normal control group, followed by the osteopenia and osteoporosis groups. In this study, a total of 1940 ASVs were obtained based on a 97% threshold. Among them, 4.4% ASVs were common among the normal control, osteopenia, and osteoporosis groups with a high number of shared ASVs observed between the osteopenia and osteoporosis groups (5.9%), as shown in Appendix A. The number of unique ASVs was higher in the osteoporosis (34.3%) and osteopenia (32.5%) groups than in the normal control group (19%).

### 3.2. Bacterial Diversity Analysis among the Experimental Groups Based on 16S rRNA Gene Signatures

The bacterial diversity and richness among the normal control, osteopenia, and osteoporosis groups were evaluated based on alpha diversity indices, including observed_ASV, Chao1, and Shannon. These alpha diversity indices revealed a high diversity and richness associated with the normal control group, followed by the osteopenia and osteoporosis groups, as shown in Figure 1. However, all of these indices showed nonsignificant differences in alpha diversity among the experimental groups based on the Kruskal–Wallis group and pairwise comparisons. Similarly, beta diversity based on the Bray–Curtis index using principal coordinate analysis revealed a lower dissimilarity in beta diversity among the three experimental groups, showing no clear separation. Additionally, the adonis function based on PARMANOVA was applied to further determine the significant difference in beta diversity, which revealed nonsignificant differences among the three experimental groups based on a *p*-value of <0.05.

### 3.3. Bacterial Community Profiling and Pattern Analysis at Taxonomic Levels in the Experimental Groups Based on 16S rRNA Gene Signatures

A total of 197 classifiable genera were identified from the 16S rRNA gene sequence by targeting the V3–V4 regions. Among them, 75 highly abundant genera were retained in each experimental group for comparison and further downstream analysis after considering a low filter count at 10% abundance in each sample. The Venn diagram (Figure 2A) showed only one unique genus associated with the normal control group, and five unique genera were only present in the osteopenia group. However, no unique genera were identified in the osteoporosis group. A total of 40 genera were shared among the normal control, osteopenia, and osteoporosis groups. Similarly, 28% of the genera were common between the osteopenia and osteoporosis groups, 6.7% genera in the normal control and osteopenia groups, and 4% were common between the normal control and osteoporosis groups. The heatmap analysis (Appendix A) revealed a distinct pattern of these predicted genera based on the relative abundance in each experimental group. Additionally, we applied a combination of heatmap and correlation analysis to determine the extent of highly dominant genera among the experimental groups. Among the top 25 genera, 13 were positively correlated among the experimental groups (Figure 2B). The majority of these bacteria, including *Dorea*, *Erysipelotrichacea*, *Streptococcus*, *Collinsella*, *Flavonifactor*, *Butyricicoccus*, *Paraprevotella*, *Parabacteriodes*, and *Colidextribacter*, showed a higher abundance in the osteoporosis group than in the healthy controls and osteopenia groups. Conversely, two genera, *Prevotella* and *Eubacterium_rumina*, were more abundantly associated with the osteopenia group, followed by that in the osteoporosis group than in healthy controls. However, all negatively correlated genera were in high abundance in healthy controls, and the majority of their abundance reduced in the osteopenia followed by osteoporosis groups.

### 3.4. Bacterial Community Shift Analysis at the Genus Level among the Experimental Groups

To further understand the significant shift of the bacterial community at the genus level among the experimental groups, we applied a two-sided Welch’s test, *p* < 0.05, as shown in Figure 3. The comparison of the normal control and osteopenia groups revealed that seven genera, namely *Blautia*, *Alloprevotella*, *Bacteriodes*, *Dorea*, unassigned, *NK4A214_group*, and *Streptococcus*, were statistically enriched between the two groups. Among them, all genera high in abundance were associated with osteopenia, except *Bacteroides*, which was high in abundance in the normal control group. Similarly, nine genera were statistically enriched between the normal control and osteoporosis groups. The majority of genera, including *Barnesiella*, *Dorea*, *Subdoligranulum*, unassigned, *Streptococcus*, *Tyzzerella*, *Oscilibacter*, and *Prevotella*, in higher abundance were associated with osteoporosis more than healthy controls. However, only *Bacteroides* abundance was high in healthy controls, and a reduced abundance was observed in the osteoporosis group. In contrast, four genera, namely *subdoligranulum*, *Collinsella*, *Desulfovibrio*, and *Flavonifractor*, were significantly enriched between the osteopenia and osteoporosis groups. However, all the genera with high abundance were associated with osteoporosis, except *Desulfovibrio*, which was in high abundance in the osteopenia group.

### 3.5. Bacterial Association and Interaction Analysis among the Experimental Groups

To identify the association and interactions within the bacterial community in osteopenia (Figure 4B) and osteoporosis (Figure 4C) with respect to healthy controls (Figure 4A), network analysis was performed, and significant positive and negative genus-level specific co-occurrence interactions were visualized. The network results revealed that bacterial interactions were higher in the osteoporosis group followed by the osteopenia group than in healthy controls. A total of 50 nodes were observed in both the osteopenia and osteoporosis groups, whereas only 29 nodes were observed in the control group, as shown in Appendix A. Among these nodes, 15 were common among the experimental groups with a higher number of unique nodes associated with osteopenia (*n* = 12), followed by osteoporosis (*n* = 9), than in healthy controls (*n* = 1). Additionally, the number of edges increased in the osteoporosis group, followed by osteopenia group, than in healthy controls, as shown in Appendix A. Moreover, eccentricity revealed a higher value of association with osteoporosis, followed by osteopenia, than in healthy controls.

NetShift analysis was used to further identify the driver genera that may cause bacterial community shifts in osteopenia and osteoporosis groups. NetShift analysis based on the comparison of the osteopenia group with the normal control group revealed that *Ruminococcaceae UCG002* was the main driver genus showing bigger red nodes (higher NESH score), followed by unidentified genera with a smaller red node (lower NESh score), as shown in Figure 5A,C. The *Ruminococcaceae UCG002* genus was positively associated with *Christensenellaceae_77_group*, *Veillonella*, and unidentified genera, whereas the unidentified genus was positively correlated with *UCG002* and *Sutterella.* Similarly, NetShift analysis based on the comparison of the osteoporosis group with the normal control group revealed three driver genera, indicated as red nodes, which were *Agathobacter*, *Clostridia_UCG014*, and *Ruminococcaceae UCG002*. Among these, *Ruminococcaceae UCG002* was the main diver (NESH score = 2.6), followed by *Agathobacter* (NESH score = 2.33), and *Clostridia_UCG014* (NESH score = 2), as shown in Figure 5B,C. *Ruminococcaceae UCG002*, the main driver genus, was positively associated with *Faecalibacterium*, *Megamonas*, and unidentified genera, whereas *Agathobacter* was positively associated with unidentified and *Roseburia* genera. Additionally, the third driver genus, *Clostridia_UCG014*, was positively associated with *Eubacteriumcoprostanoligenes_group* and the uncultured genera.

### 3.6. Predicted Functional Shift among the Experimental Groups

PICRUSt2 based on 16S rRNA gene sequences using the metacyc database revealed a total of 384 pathways associated with the experimental groups. Statistical analysis based on a *p*-value of <0.05 showed 11 predicted pathways that were significantly enriched between the normal control and osteopenia groups, as shown in Figure 6. Among them, five predicted pathways, namely PWY1G-0 (mycothiol biosynthesis), PWY-7007 (methyl ketone biosynthesis), PWY-7255 (ergothioneine biosynthesis), PWY-7098 (vanillin and vanillate degradation), and denitrification-PWY, were highly abundant in the normal control group, and their abundance reduced significantly in the osteopenia group. In contrast, six predicted pathways, namely PWY-6174 (mevalonate pathway), PWY-3661(glycine betaine degradation), PWY-6731(starch degradation), PWY-7286 [7-(3-amino-3-carboxypropyl)-wyosine biosynthesis], and methanogenesis-PWY, were more abundant in the normal control group. Comparison of normal controls and osteoporosis revealed nine statistically significant predicted pathways. Five of them, including PWY0-1296 (purine ribonucleoside degradation), PWY0-1297 (super pathway of purine deoxyribonucleoside degradation), PWY-7255 (ergothioneine biosynthesis), PWY-6397 (mycolyl-arabinogalactan-peptidoglycan complex biosynthesis), and PWY-5178 (toluene degradation), were abundant in the osteoporosis group compared to normal controls. The remaining significant pathways, including PWY-7456 (β-(1,4)-mannan degradation), PWY-7255 (ergothioneine biosynthesis), PWY-7644 (heparin degradation), and PWY-5651 (L-tryptophan degradation to 2-amino-3-carboxymuconate semialdehyde), were in high abundance in the control group, and their abundance reduced in the osteoporosis group. Additionally, ten predicted functions were statistically enriched between the osteopenia and osteoporosis groups. Among them, six predicted pathways, namely PWY-5651 (L-tryptophan degradation to 2-amino-3-carboxymuconate semialdehyde), PWY-5655 (L-tryptophan degradation), PWY-6174 (mevalonate pathway), PWY-3661 (glycine betaine degradation), PWY-7295 (L-arabinose degradation), and PWY-6713 (L-rhamnose degradation), in high abundance were associated with the osteopenia group, and reduced abundance was observed in the osteoporosis group. Conversely, the remaining three significant pathways, namely denitrification (PWY), PWY-6944 (androstenedione degradation), and PWY-6397 (mycolyl-arabinogalactan-peptidoglycan complex biosynthesis), were in higher abundance in the osteoporosis group than in the osteopenia group.

### 3.7. Correlation and Association Analyses among the Significant Taxa, Predicted Pathways, and Physiological Parameters

Pearson analysis (*p*-value of <0.05) was applied to further explore the relationship between the predicted pathways and bacterial taxa at the genus level. For better visualization, the correlated pathways and genera are shown in Figure 7A. The correlation analysis revealed that PWY-7286 and methanogenesis-PWY were significantly positively correlated with *Subdoligranulum*, *Ruminococcaceae UCG-002*, and unassigned. Similarly, PWY0-1297 and PWY0-1297 were significantly correlated with *Streptococcus*, *Flavonifractor*, and unidentified genera. PWY-7456 was significantly positively correlated with *Bacteroides*, *Desulfovibrio*, *Clostridia_UC-014*, and unidentified genera. Additionally, *Ruminococcaceae UCG-002*, *Blautia*, unassigned genus, and *Desulfovibrio* were positively correlated with PWY0-1297, PWY-51, PWY-3661, and PWY-7295, respectively. Further RDA analysis was applied to evaluate the influence of physiological parameters, including height, weight, age, and menopause period, as variables in controlling the bacterial community structure associated with reduced bone mass, as shown in Figure 7B. According to the length of the arrows, height and T-score had a strong effect on the distribution of gut microbiota, followed by the menopause period and age, compared to weight. Height and weight were strongly positively correlated with *Desulfovibrio* and unidentified genera, whereas age and T-score were strongly positively correlated with *Bacteroides*, *Agathobacter*, *Clostridia_UCS-014*, and *Ruminococcaceae UCG-002.*

## 4. Discussion

This is the first study in which we compared gut bacterial diversity, abundance, and functional pathways in Taiwanese postmenopausal women with osteopenia and osteoporosis using 16S rRNA amplicon sequencing. The bacterial diversity analysis in this study revealed that women with osteoporosis had a reduced alpha diversity, followed by those with osteopenia, as compared with that in healthy controls, which is consistent with previous reports [9,11,20], indicating that disease progression is associated with reduced microbial diversity in patients with osteopenia and osteoporosis. Microbial diversity analysis is widely considered a vital indicator of health conditions, and reduced microbial diversity is correlated with several disease progressions [29]. Recent studies have reported that dysbiosis of the gut microbiota might be a cause of imbalance reactions related to osteogenesis and osteoclasts that ultimately lead to osteopenia and osteoporosis [20,30]. Additionally, we observed significant changes in the bacterial composition and abundance at taxonomic levels in patients with osteopenia and osteoporosis compared with healthy controls. Similar changes in the composition and abundance of gut microbiota associated with osteoporosis have been reported in both human and animal studies [31]. Among the positively correlated bacterial genera, *Dorea*, *Streptococus*, and *Parabacteriodes* showed a higher abundance in the osteoporosis group, followed by the osteopenia group, compared to healthy controls, showing that these bacteria could play an important role in bone mineralization and the progression of disease. Additionally, microbial shift analysis based on Welch’s t-test (*p* < 0.05) revealed that *Blautia*, *Alloprevotella*, *Dorea*, unassigned, *NK4A214_group*, and *Streptococcus* were significantly higher in abundance in the osteopenia group than in healthy controls, indicating that these highly enriched bacteria have a strong association with the early development of bone mineralization and might cause osteopenia. Previous studies have reported that the majority of these bacteria in significantly high abundance were associated with subjects with low bone mass [32]. Similarly, *Barnesiella*, *Dorea*, *Subdoligranulum*, unassigned, *Streptococcus*, *Tyzzerella*, *Oscilibacter*, and *Prevotella* in significantly higher abundance were observed in the osteoporosis group than in healthy controls, suggesting that these key bacteria may play an important role in the development of osteoporosis. The majority of these significantly enriched bacteria with high abundance have been reported in previous studies associated with osteopenia and osteoporosis [11,22,33]. Growing evidence indicates that excessive growth of gut microbiota in both animal and human models can lead to bone loss by modeling the immune system and bone homeostasis [20,34,35]. However, the particular role of the majority of these highly enriched bacteria associated with osteopenia and osteoporosis is still unknown. Further studies are required to explore their roles using high-resolution approaches. Additionally, these highly enriched bacterial taxa associated with osteopenia and osteoporosis might be potential biomarkers for the early detection of osteopenia and osteoporosis and can aid in the treatment and prevention of osteopenia and osteoporosis.

The microbial association and interaction in a community are crucial for determining the overall structure and function of the community [17]. Network analysis is an invaluable tool to capture the association among microorganisms that maintain the balance between health and disease [36]. The co-occurrence network results revealed that bacterial interactions and associations were higher in the osteoporosis group, followed by the osteopenia group, than in healthy controls. Similarly, network eccentricity also revealed a higher value of association with osteoporosis, followed by osteopenia, than in healthy controls. These outcomes further indicated that the relationship of gut bacteria at the genus level was higher in the osteoporosis group, followed by those in the osteopenia group, than in healthy controls, suggesting a strong association in gut microbial community with disease progression. It is evident that the progression of disease changes the host physiology significantly, which, in turn, affects the pattern of microbial community interaction and association [18]. However, in such conditions, a few microorganisms, called driver microorganisms, act as key players [37]. This study further revealed that *Ruminococcaceae UCG-002* was the main driver genus that played key roles in changing microbial associations and interactions associated with osteopenia, whereas two other diver genera, *Agathobacter* and *Clostridia_UCG014*, along with *Ruminococcaceae UCG-002*, were also found to be responsible for changing the bacterial community in osteoporosis. These findings suggest that these driver bacteria can play a key role in the initiation and progression of osteopenia and osteoporosis. However, the biological significance of these bacteria in the milieu of integrative bacterial community development is yet to be further explored. The genus *Ruminococcaceae UCG-002* belongs to the family *Ruminococcaceae* and has been previously reported to be associated with osteopenia and osteoporosis. Previous studies have reported that the abundance of *Ruminococcaceae* associated with animal and human models increased following calcium supplementation [38,39]. *Agathobacteria* is a new genus that belongs to the family *Lachnospiraceae* [40] and was recently reported to be an abundant genus along with other bacteria in both primary osteopenia and healthy controls [20]. The primary metabolite produced by this genus is butyrate, which is a short-chain fatty acid (SCFA) [41]. Many studies have indicated that SCFAs help in bone formation by increasing the production of steopontine and sialoprotein [42,43]. Additionally, *Clostridium* usually initiates the accumulation of Tregs, which are considered inhibitors of osteoclast differentiation in the lamina propria of the colon [44]. The absence of *Clostridium* strains causes a reduction in Foxp3 Treg levels with an increase in bone loss [44,45].

Recently, growing evidence has shown that gut microbiota can induce bone remodeling through metabolic pathways in both animal and human models [20,46]. In this study, microbial functional prediction based on 16S rRNA sequencing revealed several pathways that were significantly enriched in osteopenia and osteoporosis groups compared with healthy controls. Three highly abundant pathways, namely mycothiol biosynthesis, methyl ketone biosynthesis, and ergothioneine biosynthesis, were significantly reduced in the osteopenia group compared with healthy controls, whereas the other three abundant pathways (mevalonate pathway, glycine betaine degradation, and starch degradation) were significantly high in abundance in the osteopenia group compared to the healthy controls. Bacteria use mycothiol as a protectant to detoxify reactive nitrogen and oxygen species [47]. Previous studies have reported that reactive nitrogen and oxygen species are vital for inducing various signaling events, such as intracellular Ca^2+^ levels, regulation of mitogen-activated protein kinases (MAPKs), and transcription factors that are involved in bone formation [48,49]. Similarly, ergothioneine is synthesized by various microorganisms and is believed to protect cells from oxidative stress. Therefore, a reduced abundance of these pathways compared with those in healthy controls might be a primary cause of osteopenia. Previously, it has been reported that the mevalonate pathway is involved in cholesterol biosynthesis, a vital regulatory pathway related to bone remodeling. Additionally, the mevalonate pathway produces geranylgeranyl and farnesyl pyrophosphates, which are crucial for protein prenylation [50]. High production of protein prenylation favors bone resorption over bone formation [51,52]. Additionally, glycine betaine degradation pathways have also been reported to be involved in low bone mass [53]. Compared with healthy controls, the osteoporosis group showed a higher abundance of purine ribonucleoside degradation and super pathway of purine deoxyribonucleoside degradation. Both of these pathways are related to purine degradation. Previous studies have also reported that purine degradation is associated with low bone mass. Correlation analysis revealed that most of these significantly enriched pathways were positively correlated with statistically enriched bacteria at the genus level, which were previously found to be statistically enriched in osteopenia and osteoporosis groups [9,54,55]. These significantly enriched pathways and bacteria and their positive correlation indicate that they are potentially involved in the initiation and progression of osteopenia and osteoporosis. However, these significant differential pathways need to be further validated using metabolomics approaches and reliable quantitative methods. Additionally, further studies are warranted to determine the relationship between these highly correlated pathways and their significant bacteria using quantitative approaches. The association between physiological parameters and gut microbial community is complex. However, previous studies have reported that these parameters, including BMI and BMD, are involved in the alteration of gut microbial community structure [23]. In this study, body height and T-score were the primary factors controlling the gut microbial community structure, which is in line with previous reports [56,57]. Height and weight were strongly positively correlated with *Desulfovibrio* and unidentified genera, whereas age and T-score were strongly positively correlated with *Bacteroides*, *Agathobacter*, *Clostridia_UCS-014*, and *Ruminococcaceae UCG-002. Desulfovibrio* belongs to the H_2_S-producing *Desulfovibrionaceae* family and was previously reported to be associated with weight loss [57]. Additionally, the correlation of *Agathobacter*, *Clostridia_UCS-014*, and *Ruminococcaceae UCG-002* with age and T-score is in line with our analysis of driver microbes in which these three bacteria were responsible for changing the microbial diversity structure. We believe these three driver bacteria might be the key players in the initiation and progression of osteopenia and osteoporosis in Taiwanese postmenopausal women. However, further studies are required to explore this association at a laboratory scale.

In conclusion, the results of this study revealed that the gut microbial community interaction is a crucial factor in the initiation and progression of osteopenia and osteoporosis in postmenopausal women. Additionally, this is the first time we highlighted the driver bacteria responsible for bacterial community shift, which might be the key players in the initiation and progression of osteopenia and osteoporosis in postmenopausal women. Overall, this study suggests that microbial community interaction is more important than the taxonomic differences in knowing the critical role of gut microbiota in postmenopausal women associated with osteopenia and osteoporosis. However, we did not consider diet patterns, which may interfere with the composition and structure of the gut microbiome. Additionally, this study was based on 16S rRNA sequencing, which does not have sufficient resolution depth to identify the bacteria at the species level. Therefore, in future studies, a high-resolution approach targeting long reads of 16S rRNA must be considered to identify the bacteria at the species level.

## Figures and Tables

**Figure 1 microorganisms-11-00234-f001:**
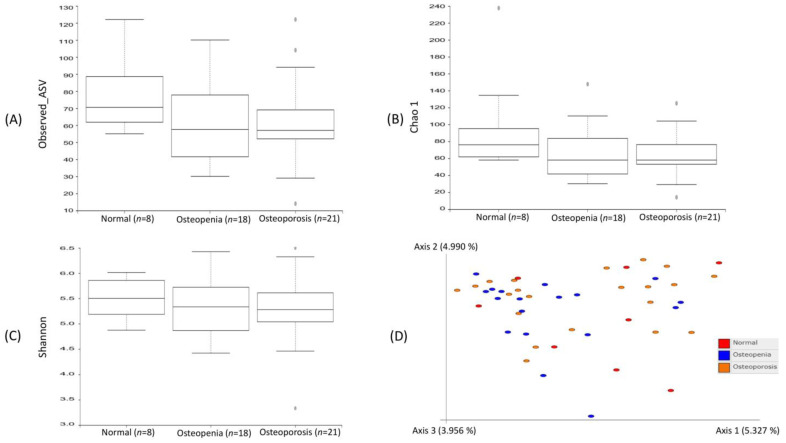
Comparison of bacterial community diversity among the experimental groups: normal control, osteopenia, and osteoporosis. Alpha diversity among the experimental groups was measured by Observed_ASV, Chao1, and Shannon, whereas beta diversity based on Bray–Curtis was measured using principal coordinate analysis.

**Figure 2 microorganisms-11-00234-f002:**
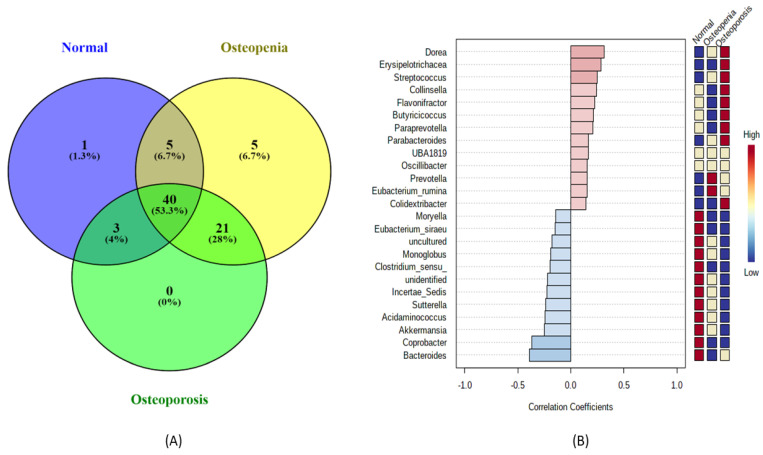
Comparison of bacterial community composition and abundance associated with experimental groups: normal control, osteopenia, and osteoporosis. Venn diagram (**A**) representing the shared and unique genera. Correlation analysis (**B**) showing the positive correlation (red bars) and negative correlation (blue bars) with a heatmap that denotes the pattern or shift of genus among the experimental groups.

**Figure 3 microorganisms-11-00234-f003:**
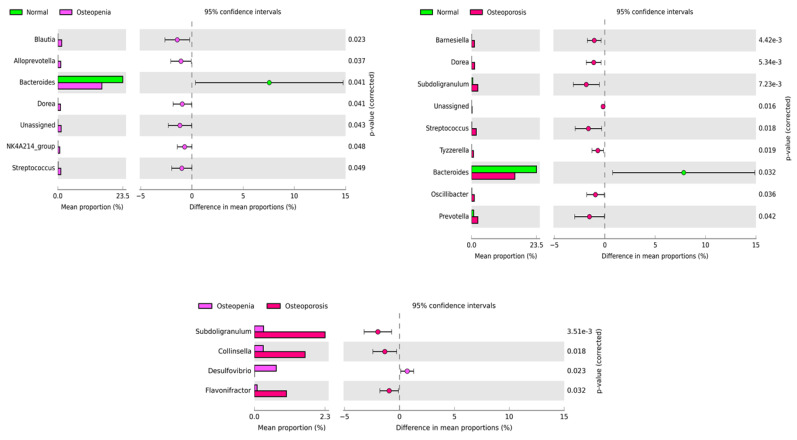
Enriched bacterial genera based on 16S rRNA among the experimental group. The left panel of each figure shows the bacterial abundance of differentially enriched genera. The right panel represents the significant difference at *p* < 0.05. The middle panel indicates the mean proportion of differentially enriched bacterial genera at a 95% confidence interval.

**Figure 4 microorganisms-11-00234-f004:**
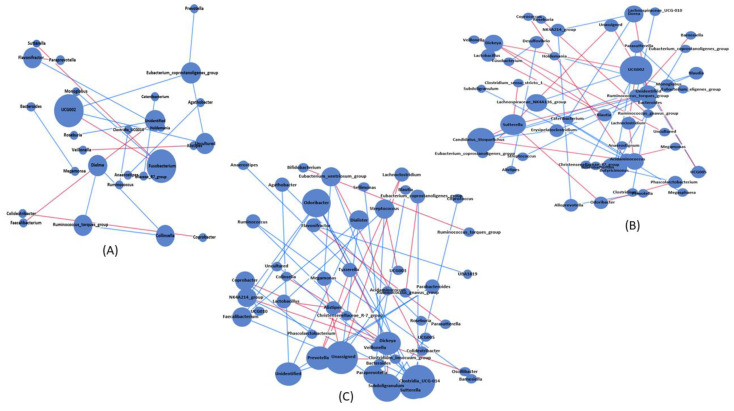
Genus-level bacterial association network analysis based on spearman among healthy controls (**A**), osteopenia (**B**), and osteoporosis (**C**). Nodes in blue color represent significant bacterial genera and lines connecting these genera indicate edges. The size of nodes represents interconnectedness among the genera. Lines with red color indicate positive correlations and red lines represent negative correlation.

**Figure 5 microorganisms-11-00234-f005:**
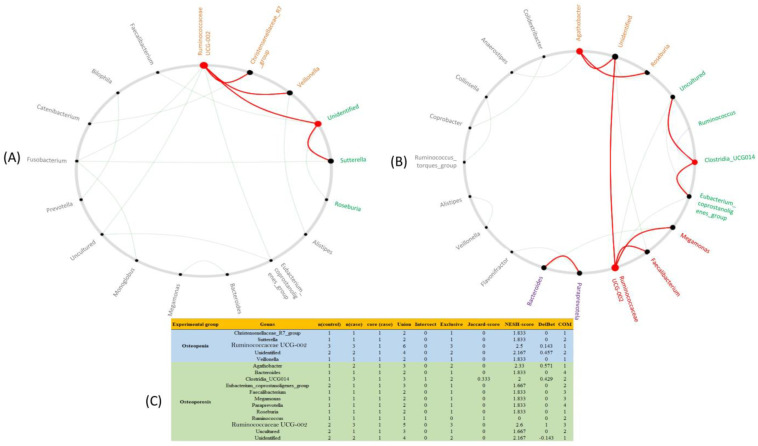
The driving genera that caused bacterial shift between the osteopenia (**A**) and osteoporosis (**B**) groups. The red color nodes denote the driver genera identified using the NetShift technique. The node sizes represent the NESH scores, and the characteristics of these nodes between osteopenia and osteoporosis groups are provided in table (**C**).

**Figure 6 microorganisms-11-00234-f006:**
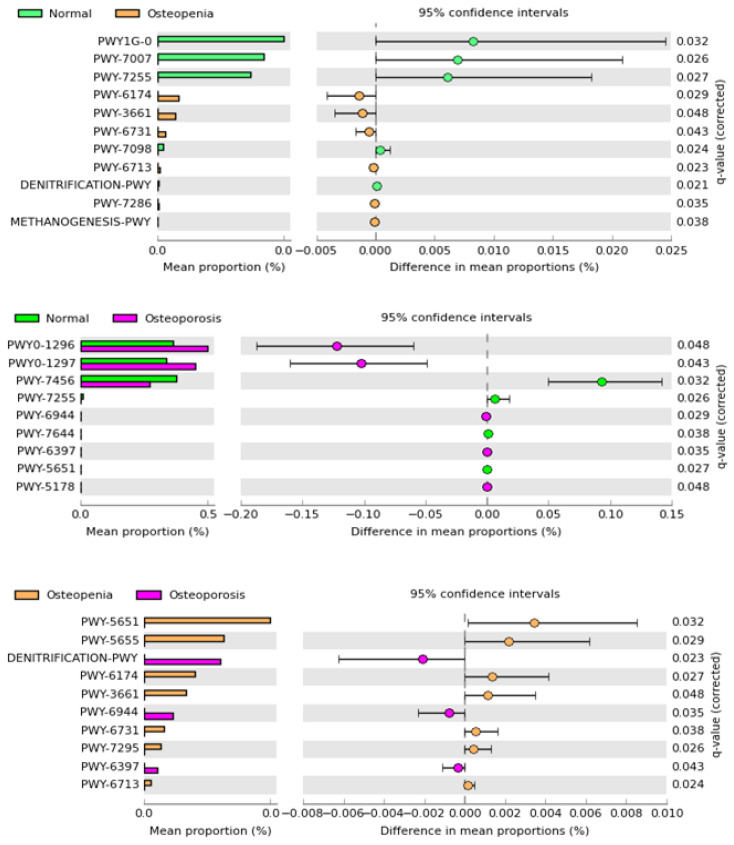
The post hoc plot of enriched microbial predicted functions among three health conditions (healthy, polyps, and cancer). The left panel of these figures shows the abundance ratio of differentially enriched KEGG pathways. The right panel represents the significant difference at *p* < 0.05, whereas the middle one indicates the mean proportion of differentially enriched KEGG pathways in the 95% confidence.

**Figure 7 microorganisms-11-00234-f007:**
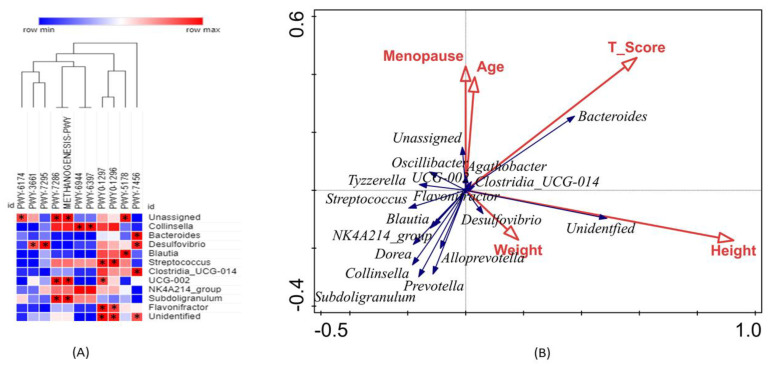
Pearson correlation analysis between the bacterial gut community at genus level and predicted pathways based on 16S rRNA amplicon (**A**). The positive and negative correlations are indicated in red and green colors, respectively, with significance at *p* < 0.05 as indicated *. Redundancy analysis (**B**) shows the relationship between the bacterial taxa (blue arrows) and physiological variables (red arrows).

## Data Availability

The data presented in this study are available on request from the corresponding author.

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
