# Peer review of "Inferring Bacterial Community Interactions and Functionalities Associated with Osteopenia and Osteoporosis in Taiwanese Postmenopausal Women"

_microorganisms, 2023, doi:10.3390/microorganisms11020234_

Round 1
Reviewer 1 Report
This manuscript investigates the gut bacterial community interactions along with their composition and functional changes in healthy postmenopausal women as compared to osteopenic and osteoporotic ones in Taiwan. The authors report that enriched bacterial genera were more abundant in osteoporotic/osteopenic patients that in the healthy controls and were also presented with a higher amount of interactions. Furthermore, they found that most of these bacterial populations were associated with important metabolic pathways and were correlated with the height and the T-score.
This is an interesting work and the authors used efficiently a variety of bioinformatic tools to produce the results and draw the conclusions. There are some minor points that need to be addressed.
· It seems that there is something missing at the end of the abstract as it stops unexpectedly.
· Line 68, put 13 and 14 as superscripts.
· Line 94. It is crucial to point out that this paper is a descriptive one and does not prove that these changes in the bacterial populations result in the development of osteopenia or osteoporosis. Therefore, the authors need to pay attention to the relevant parts throughout the entire manuscript. For example, the wording “may lead to osteopenia and osteoporosis in postmenopausal Taiwanese women” in line 94 needs to be rephrased.
· Line 194, what do the authors mean by “good quality sequence reads”? In addition, what was the cut-off for the reads?
· The heatmap in FigS2 needs to be removed in the main manuscript. Importantly, it is essential to show all the samples in the heatmap and how they cluster together. This form with 3 groups cannot be accepted, it is important to show the distribution as well.
· Line 369, please correct “low bone biomass”.
· In section 3.7, why did the authors analyse the correlations of height and weight and not BMI?
Author Response
We are grateful to the reviewer for his/her insightful comments on our paper. We have been able to include changes to reflect most of the suggestions provided by the reviewers.
Reviewer 2 Report
The authors studied the relationship between osteopenia and osteoporosis and gut bacterial ‘s community interactions and functionalities. Network-based analysis was applied to identify the key genera that are distinct between two groups (control, osteopenia and osteoporosis). Although there are many limitations, this work is useful for people to understand the disease progression of osteopenia and osteoporosis. Considering osteoporosis is more severe than osteopenia, could the authors
(1) discuss the difference between the two groups?
(2) The change of some genera in term of composition, interaction, and functionality reflects the disease progress from normal to osteopenia and then to osteoporosis?
Author Response

(The authors gave the same response as above.)

Round 2
Reviewer 2 Report
Thank the authors for revising the manuscript according to my suggestion. The manuscript looks good to me, and I support the publication of this work.